# Prokaryote Composition and Structure of Rumen Fluid before and after In Vitro Rumen Fermentation

**Rajan Dhakal** [1,*]**, André Luis Alves Neves** [1,*]**, Rumakanta Sapkota** [2]**, Prabhat Khanal** [3]
**and Hanne Helene Hansen** [1]

1 Department of Veterinary and Animal Sciences, Production, Nutrition and Health, University of Copenhagen, Grønnegårdsvej 15, DK-1870 Frederiksberg C, Denmark; hhh@sund.ku.dk
2 Department of Environmental Science, Aarhus University, DK-4000 Roskilde, Denmark; rs@envs.au.dk
3 Faculty of Biosciences and Aquaculture, Nord University, Skolegata 22, 7713 Steinkjer, Norway; prabhat.khanal@nord.no
* Correspondence: dbk638@ku.dk (R.D.); andre.neves@sund.ku.dk (A.L.A.N.)

**Abstract:** Background: This study aimed to investigate the impact of in vitro rumen fermentation (IVRF) on the microbiome structure and composition of rumen fluid before and after fermentation assays. Methods and Results: Six separate fermentation batches were run for 48 h using maize silage as the basal feed. Rumen fluid samples were analyzed before (RF; only rumen fluid inoculant) and after 48 h fermentation assay (MS; maize silage as the substrate) and further processed for microbiome analysis using amplicon sequencing targeting the V4 region of the bacterial 16S rRNA gene. Bacterial alpha diversity revealed that the Shannon index and observed index were similar between MS and RF fluid. The core microbiome was detected in 88.6% of the amplicon sequence variants in MS and RF. Taxonomic analysis at the phylum level showed similar abundances of *Bacteroidetes*, *Proteobacteria*, *Firmicutes*, *Verrucomicrobiota*, *Spirochaetota*, *Patescibacteria*, and *Campilobacterota* in MS and RF. The Bray–Curtis distance matrix showed similar bacterial community structure among MS and RF samples. Conclusion: Our results indicated that the in vitro procedure did not affect the bacterial community structure compared to the original rumen fluid inoculum. It should be noted that assessing the microbiome at a single endpoint (i.e., 48 h) may not provide a comprehensive understanding of the microbiome profile dynamics. However, the findings of this study provide a basis for future microbiome-based in vitro fermentation tests and confirm that the technique allows a high degree of species diversity that approximates the rumen function in vivo.

**Keywords:** rumen microbiome; *Bacteroidota*; *Proteobacteria*; *Firmicutes*; 16S rRNA

## 1. Background

Feeding is an essential component of any ruminant production system, playing an important role in the long-term profitability and sustainability of the business. The chemical composition, anti-nutritional components, and digestibility of feedstuffs used for ruminants must be determined regularly to ensure the success of the enterprise both financially and environmentally. The quality of the feed used in ruminant operations can be examined in three possible ways: (1) testing the feed with live animals (in vivo), (2) performing chemical analysis of the feed [1], or (3) undertaking in vitro feed evaluations [2]. In vivo testing of feed is desirable and is considered the gold standard. However, it is costly and labor intensive and may impact animal health and well-being during the experiment. Chemical analysis of the feed is useful, but it does not provide information on how the feed interacts with the digestive tract, growth, performance, and well-being of the animal. In vitro feed evaluation is a viable alternative to the first two options because it simulates a specific segment of the digestive tract of animals [3] and is a cost-effective, non-invasive, easy-to-operate, and efficient method for quickly assessing the nutritional content of feedstuffs

without the need for using live animal trials. However, in vivo studies should confirm the in vitro findings.

The in vitro method is used to simulate the rumen in vivo and can be broadly classified into three categories: dynamic, semi-dynamic, and static systems (batch cultures). In each case, a sample feed is incubated with rumen fluid and a buffer in a controlled setting, and a predetermined time interval is used to maintain the ideal physico-chemical properties in the test flask for the rumen microbes to thrive [4]. In a static system, in vitro rumen fermentation is carried out by combining a sample of feed mixed with rumen fluid and a buffer made of macro and micro minerals. In a static in vitro system, the experimental conditions remain constant throughout the duration of the experiment. This means that there is no flow or movement of buffer or feedstuff in the fermentation vessel once the experiment begins. A semi-dynamic in vitro system combines aspects of both static and dynamic systems. The dynamic and semi-dynamic systems adhere to the same principles as the static system, but they enable the influx and outflux of buffer and waste and allow the expulsion of accumulated gas at a predetermined pressure or rate. In semi-dynamic systems, the experimental conditions will change periodically, but there is no continuous flow of feed stuff. Furthermore, the dynamic in vitro systems allow the continuous flow of feedstuff through the fermentation vessel. However, it is important to note that the semi-dynamic system may not provide all the functionalities of the dynamic system. For all three systems, the feeds are incubated under anaerobic conditions in a fermentation vessel, or glass bottles equipped with an automatic gas measurement system [5], fermentation tubes, or glass syringes [3]. The choice of system depends on the specific research goals and the biological processes to be studied. The fermentation process is typically carried out by maintaining a temperature of approximately 39 °C and a pH of around 6.5–7.0, similar to the conditions in the rumen in vivo. Based on research interest and objectives, fermentation is carried out for a period of 24–96 h [6].

In vitro rumen fermentation can be used to study rumen function, including the degradation of feedstuffs, fermentation kinetics, the production of volatile fatty acids (VFAs), total gas production, the concentration and yield of methane in total gas, the activity of rumen microorganisms, and the effects of feed additives on microorganisms [7–10]. Fermentation is influenced by the feed type and inoculum source (rumen fluid) and having the same microbial composition before and after the fermentation is desirable. However, there is limited knowledge of the microbiome composition before and after a fermentation assay. We hypothesized that the procedures used during in vitro fermentation do not affect the rumen prokaryotic (bacteria and archaea) community structure and that this prokaryotic composition remains unchanged compared to that of the original rumen fluid even after a 48 h fermentation. In this work, we aimed to characterize the rumen prokaryotic profile in pre- and post-fermentation samples to verify our hypothesis.

## 2. Materials and Methods

### 2.1. In Vitro Fermentation

Maize silage was used as the basal substrate for the fermented samples and 500 mg were weighed in 100 mL Duran® bottles. Fermentations were undertaken for 48 h with triplicates in each trial for the samples incubated with maize silage. Two rumen-cannulated heifers at the Large Animal Hospital from the University of Copenhagen (Taastrup, Denmark) were donors of rumen fluid (solid and semi-solid phases) for each fermentation. The heifers fasted for 12 h before rumen fluid collection. The cannulated animal use was authorized by Danish law under the research animal license no. 2012-15-2934-00648. Animals were fed haylage (85% dry matter, 7.5 MJ/kg metabolizable energy, and 11% crude protein) ad libitum for over six weeks before the experiment. Six in vitro fermentation assays were undertaken as described by Dhakal et al. [11]. Briefly, a four-part buffer solution was prepared and flushed with $CO_2$ for two hours to ensure anaerobic conditions, and the temperature of the medium was maintained at 39 °C before the addition of the rumen fluid. A reduction agent composed of sodium sulfide and sodium hydroxide was added

10 min before the addition of the rumen fluid. An equal amount of rumen fluid from each cannulated heifer was filtered through a double layer of commercial cheesecloth before being added to the buffered media. This rumen fluid was added to the buffer at a 1:2 ratio, and 90 mL of the rumen–buffer inoculum was added to each bottle, flushed with $CO_2$, and capped with an ANKOM$^{RF}$ (ANKOMRF Technology, Macedon, NY, USA) module. Each ANKOM$^{RF}$ module is an automated system with a pressure sensor (pressure range: from −69 to +3447 kPa; resolution: 0.27 kPa; accuracy: ±0.1% of measured value) and radio frequency that sends and receives signals to computer software via a base station. The recording time and global release pressure for valve opening were set to 10 min and 0.75 PSI, respectively.

To analyze the microbial composition and structure of in vivo rumen fluid (fresh inoculant, 0 h, without added maize silage), 10 mL of rumen fluid samples ($n$ = 6)) were collected and stored at −20° before each incubation. Similarly, after the end of each fermentation, rumen fluid samples incubated for 48 h with maize silage ($n$ = 9) were collected and stored at −20° until further analysis.

### 2.2. DNA Extraction

Frozen samples were thawed and 2 mL of rumen fluid sample from each bottle was transferred into a new sterile tube and centrifuged at $15,000 \times g$ for 10 min to obtain cell-rich pellets for genomic DNA extraction. Following the manufacturer's protocols, DNA from the cell-rich pellets was extracted using the Bead-Beat Micro Ax Gravity (A&A Biotechnology, Gdynia, Poland). The concentration and purity of the extracted DNA were tested with a NanoDrop Lite UV-Vis spectrophotometer (Thermo Fisher Scientific, Waltham, MA, USA).

The bacterial primers 515F (GTGCCAGCMGCCGCGGTAA) and 806R (GGACTACHV-GGGTWTCTAAT) along with the Illumina Nextera overhang adapters were used to amplify the V4 region of the bacterial 16S rRNA region [12]. For the first PCR, thermocycler conditions were 95 °C for 2 min, 33 cycles of 95 °C for 15 s, 55 °C for 15 s, 68 °C for 40 s, and the final elongation at 68 °C for 4 min (SimpliAmp Thermal Cycler, Applied Bio-systems, Newton Dr, Carlsbad, CA, USA). Each PCR reaction of 25 µL consisted of 5xPCRBIO HiFi Buffer (5 µL) (PCRBiosystems, UK), 2 ng of DNA template, 0.25 units of PCRBIO HiFi Polymerase (PCRBiosystems, UK), 0.5 mM of forward and reverse primers, 0.5 µL bovine serum, and 16.25 µL H20. Gel electrophoresis was run to check the amplification in each sample.

After PCR1, a second PCR (PCR2) was performed to add unique index combinations (i7and i5) and adaptors. For PCR2, thermocycler conditions were 98 °C for 1 min, 13 cycles of 98 °C for 10 s, 55 °C for 20 s, 68 °C for 40 s, and the final elongation at 68 °C for 5 min. Subsequently, the amplicon product was cleaned using HighPrep™ magnetic beads (MagBio Genomics Inc., Gaithersburg, MD, USA), according to the manufacturer's instructions. Then, gel electrophoresis was run for the first and second PCR products of each sample to confirm the results. Finally, amplicons were pooled in equimolar concentration, and sequencing was carried out using the Illumina MiSeq platform.

### 2.3. Bioinformatics

Using the DADA2 plugin [13] implemented in the software QIIME2, the DNA reads acquired from the Illumina MiSeq run were analyzed [14]. The 'dada2 denoise-paired' command was used to screen for chimeras after denoising, joining, dereplicating, trimming primers (forward and reverse), and building amplicon sequence variant (ASV) tables. ASVs were then given a taxonomic classification using the SILVA 138 database and the 'feature-classifier classify-sklearn' method [15]. The taxonomy files and ASV table were analyzed using R version 4.2.1 [16] for data analysis and visualization. The *vegan* (version 2.6-4) and the *phyloseq* (version 1.40.0) packages, by Oksanen et al. [17] and McMurdie and Holmes [18], were used to run the diversity-based analysis. The linear discriminant analysis effect size (LEfSe) using LDA > 2 was used to identify differentially abundant bacteria and archaea among the two groups using the R package *microbiomeMarker* [19].

Alpha diversity was estimated using richness (observed) and the Shannon diversity index. The significance of differences in relative abundance of prokaryotic phyla and alpha diversity between RF and MS incubation was evaluated using the Kruskal–Wallis test. For beta diversity-based calculations, the ASV table was transformed to relative abundance and a Bray–Curtis distance matrix was used for visualization using principal coordinate analysis (PCoA). For partitioning of variance, distance matrices were subjected to permutation analysis of variance (PERMANOVA) using the *adonis* test from the *vegan* package.

## 3. Results

The amplicon sequencing generated 1,844,731 total reads giving an average of 23,351 ± 8562 reads per sample. Following quality filtration and the removal of chloroplast and mitochondrial reads, the sequence count was reduced to 1,844,661. The final average number of counts per sample that were assigned to ASVs (post-filtering) was 23,350 ± 8561.

As shown in Figure 1a, no difference ($p > 0.05$) was observed in alpha diversity metrics using the Shannon index (MS: 4.79, RF 4.77). Likewise, no differences of species richness (observed) (MS: 323.39, RF: 312.30) between rumen fluid fermented with maize silage (MS) and rumen fluid inoculum (RF) were observed. The first column in the UpSet plot shows the commonality of ASVs in the two rumen fluid samples while the second and third show the unique ASV. A total of 420 ASVs (88.6%) were shared between MS and RF (core microbiome), as shown in Figure 1b. The MS had 48 (10.13%) unique ASVs and RF had 6 (1.27%) unique ASVs.

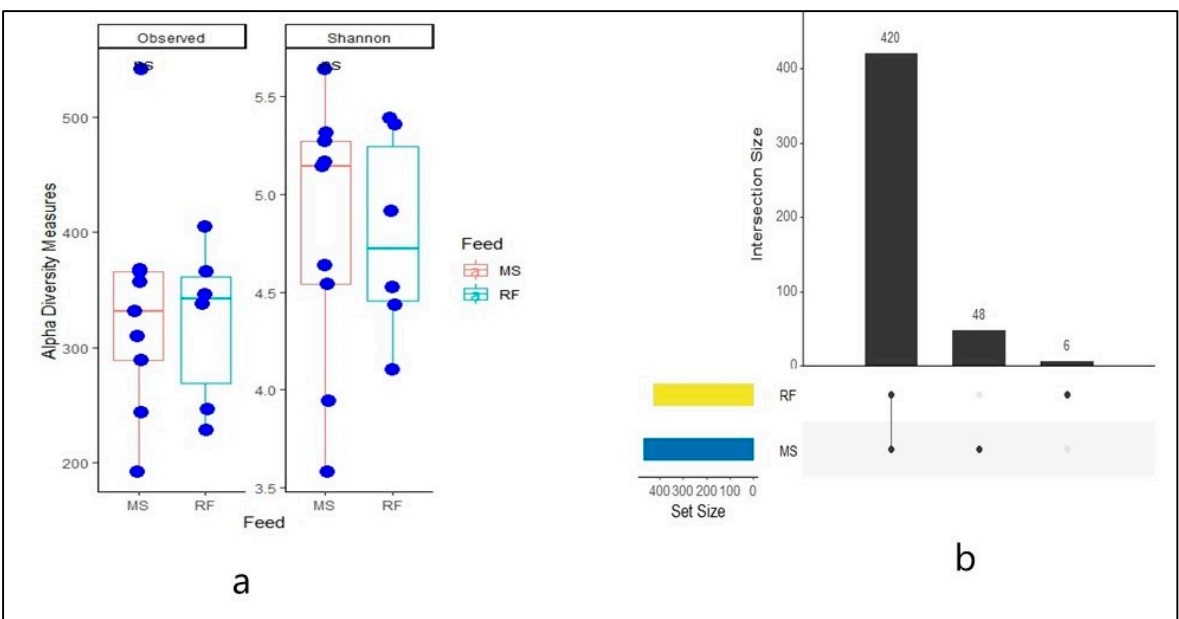

**Figure 1.** Box plot of alpha diversity metrics using observed (richness) and Shannon diversity index (**a**) and UPset plot (**b**) of number of ASVs common in rumen fluid (RF) and maize silage (MS).

The taxonomic analysis at the phylum level did not show statistical difference ($p > 0.05$) between RF and MS (Figure 2a). The top eight phylum were Bacteroidota (MS: 45.67%, RF: 48.52%), Proteobacteria (MS: 17.56%, RF: 18.87%), Firmicutes (MS: 17.18%, RF: 13.41%), Verrucomicrobiota (MS: 7.89%, RF: 8.78%), Spirochaetota (MS: 5.80%, RF: 4.68%), Patescibacteria (MS: 2.92%, RF: 2.36%), Campilobacterota (MS: 0.86%, RF: 0.14%), and Cyanobacteria (MS: 0.38%, RF: 1.8%). Figure 2b shows the microbiome community structure based on the fermentation batch and pre- (MS) and post-incubation (RF). PERMANOVA analysis revealed no significant difference between batches regarding the bacterial structures ($p > 0.05$). The abundance at the phylum level was similar ($p > 0.05$) between the MS and RF, as shown in Figure 3, except for Cyanobacteria ($p < 0.05$). To better understand the

dominance of specific bacteria within the two groups, we used the LEfSe method (Figure 4). The genera Prevotella, Gastranaerophilales, Rhodospirillales, and Burkholderiales were more abundant in RF and the genera Methanobrevibacter, RF39, Ruminiclostridium, [Eubacterium]_ventriosum_group, Desulfovibrio, Mogibacterium, Veillonellaceae_UCG−001, Schwartzia, and Lachnospiraceae_UCG-004 were more abundant in MS.

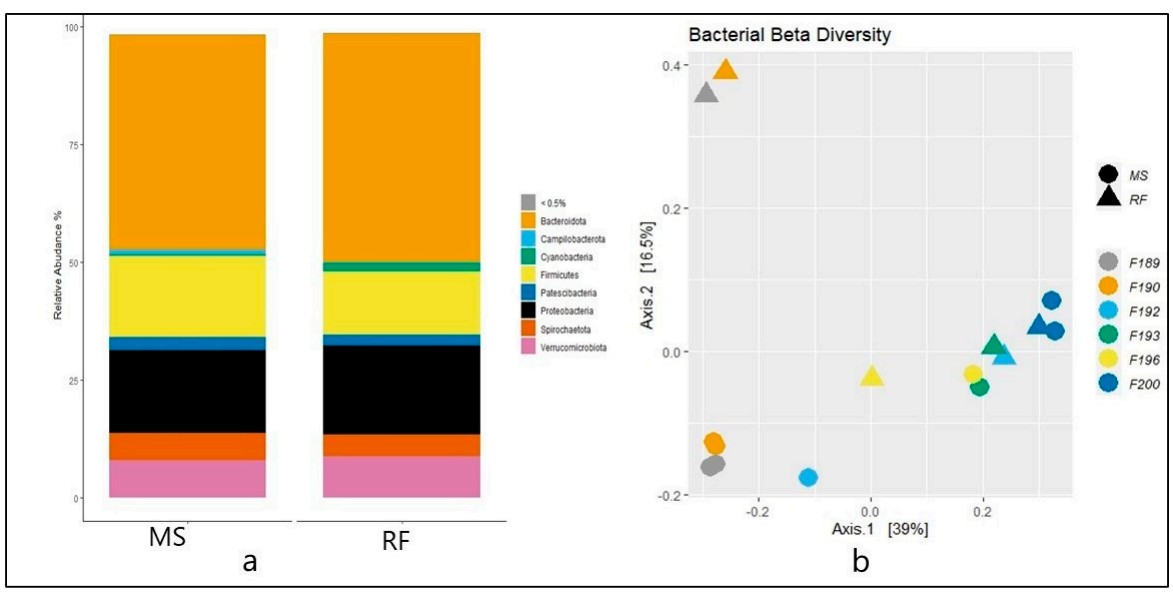

**Figure 2.** (**a**) The relative abundance (**a**) and principal coordinate analysis of Bray–Curtis distance beta diversity (**b**) of prokaryote phylum in rumen fluid (RF) and filtrate from in vitro rumen fluid fermented with maize silage (MS). Individual fermentation batches are shown with batch numbers and colored dots in 2b.

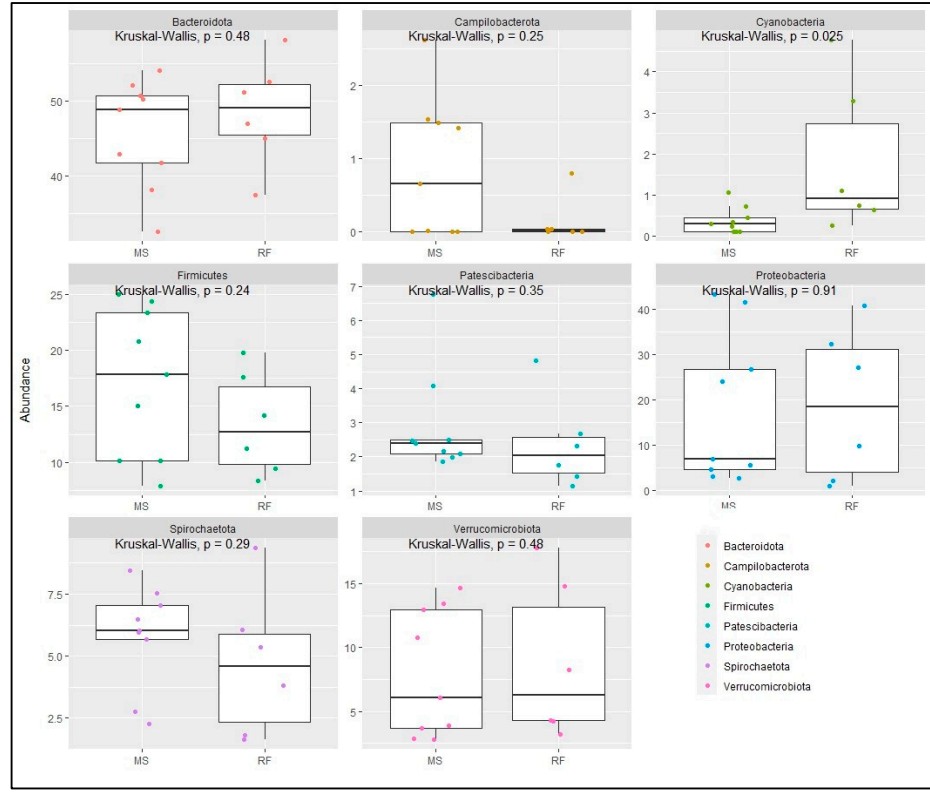

**Figure 3.** Relative abundance of a selected phyla of bacteria from in vitro fermented maize silage (MS) and rumen fluid (RF).

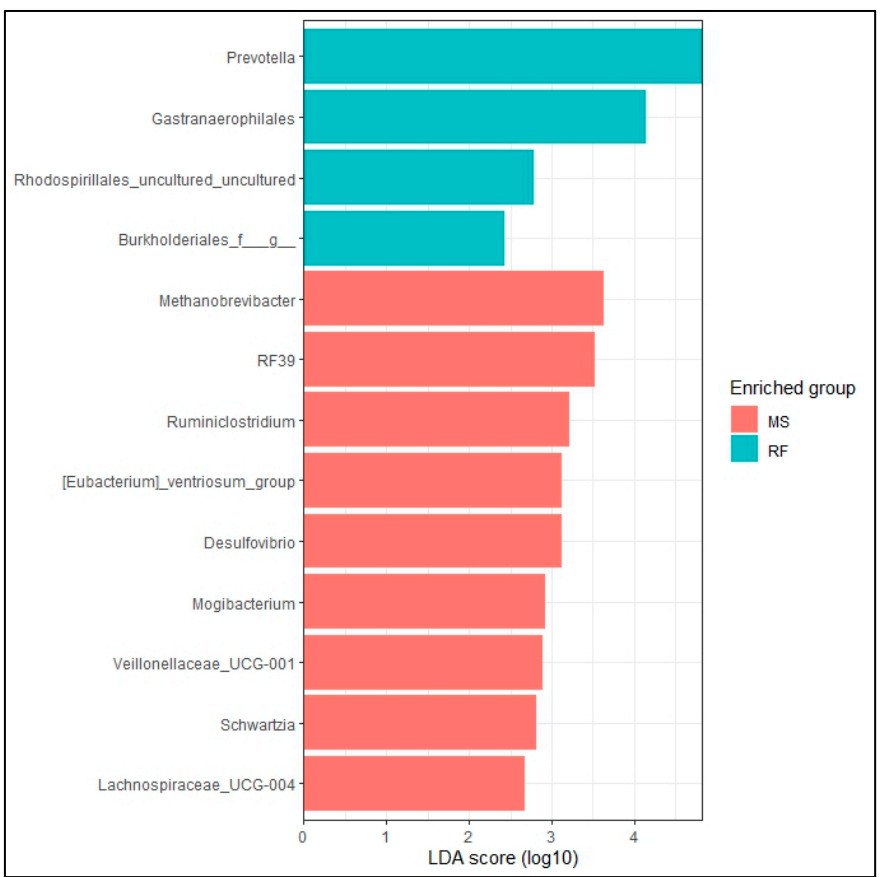

**Figure 4.** LEfSe analysis of rumen microbiome at genera level of prokaryote microbes. Differentially abundant (*p* < 0.05, LDA > 2) genera in rumen fluid (RF) and filtrate from in vitro rumen fluid fermented with maize silage (MS) are shown in the horizontal bars with their respective LDA score.

## 4. Discussion

This study examined the microbial composition of fresh rumen fluid (0 h, without addition of maize silage) and filtrate from the post-48 h fermentation stage of maize silage. We used maize silage as substrate because it is regarded as one of the most important feedstuff materials used in ruminant feeding. To our knowledge, there are no specific comparisons of microbial community structure pre- and post-fermentation. The analysis of the microbial diversity of rumen fluid (inoculant) and maize silage fermented for 48 h showed similarity in community structure and composition before and after the in vitro fermentation. In this study, the most abundant phyla were *Bacteroidota*, *Proteobacteria*, *Firmicutes*, *Verrucomicrobiota*, *Spirochaetota*, *Patescibacteria*, *Campilobacterota*, and *Cyanobacteria*, which is similar to earlier reports by Wei et al. [20] and Terry et al. [21]. The rumen microbiome consists mainly of cellulolytic and amylolytic taxa and microbes that utilize fermentation byproducts like methanogens [22]. The phyla *Bacteroidota*, *Proteobacteria*, and *Firmicutes* are responsible for fiber breakdown in the rumen [23]. The LeFSe and relative abundance analysis are in agreement. The abundant genera of MS fall within the phyla *Firmicutes*, *Desulfobacterota*, and *Euryarchaeota* and similarly abundant genera of RF fall within the phyla *Bacteroidota*, *Cyanobacteria,* and *Proteobacteria.* This study demonstrates that the in vitro system effectively replicates the in vivo rumen environment, as demonstrated by the comparable phyla levels that were observed before and after fermentation. This validation confirms the reliability of this methodology for studying the rumen microbiota.

The source of rumen fluid and replication of the in vivo conditions in in vitro fermentation are the key components to overcome the challenging steps involved in in vitro rumen fermentation. The source of rumen fluid can be from live animals fed production [24] or maintenance rations [7], rumen fluid from slaughterhouses [25], or frozen

rumen fluid [26,27]. Henderson et al. [28] reported that the core microbiome in vivo is conserved across geographical regions no matter the species or feed; however, the relative abundances of the microbes vary with the diet and the ruminant species. Hence, to replicate in vivo conditions during in vitro fermentation, it is essential to maintain donor animals, their diets, and the basal feed in the bottle constant throughout the experiments to minimize any variations in the microbial community and structure.

We found similarities in the overall bacterial community estimated by the Bray–Curtis index between two fluid types. Ma et al. [26] observed similar results when fresh and frozen rumen fluids were studied for microbial community profiling. It was expected that the DNA of the inactive or dead microorganisms were retained in the fermentation flask in our research. After incubation, we assumed that the overall (most abundant) microorganisms remained active in both the inoculation source and the flask. This is supported by the lack of difference in alpha diversity metrics in both richness and evenness between MS and RF. However, any significant difference in the proportion of readily fermentable carbohydrates as substrates in the in vitro incubation would have an impact on the amylolytic microbial population that grows at a faster rate than the fibrolytic microbes, but this was not observed in the present study.

Although there was no change in the overall microbial diversity between RF and MS, the in vitro procedure affected the growth of *Cyanobacteria*. This phylum is commonly found in soil and water and its relative abundance in the rumen is less than 1% [29]. *Cyanobacteria* are aerobic microbes that can convert inert nitrogen into the organic form, but they can utilize carbohydrates in a deficient $N_2$ concentration [30]. The absorption of oxygen from drinking water as well as contact with the air during the sample collection may have favored the growth of these bacteria in the rumen fluid. Neves et al. [31] reported that the presence of these microbes could be related to $O_2$ scavenging and sugar fermentation under the restricted aerobic conditions in the rumen.

Having the same inoculum source (animals) is the goal in every in vitro rumen fermentation. However, the donor animals undergo different physiological and nutritional changes that can affect the microbiome profile [32–34] of the inoculum. Such physiological and nutritional changes make in vitro rumen fermentation difficult to replicate [34]. This research shows that these challenges can be overcome by using the same basal treatment and the same inoculum source in every fermentation batch, followed by a comparison of the microbiome of the basal treatment (control) in each batch. Such a comparison provides insight into fermentation characteristics and whether there are any differences in the microbial populations of in vivo and in vitro rumen fluid.

## 5. Conclusions

The study indicates that the rumen prokaryotic microbial structure and composition in both pre- and post-in vitro fermentation samples remained relatively stable despite a 48 h incubation period. This composition in the in vitro filtrate after incubation did not significantly differ from the rumen fluid before incubation (inoculum). Nevertheless, the study demonstrates that the rumen prokaryotic population remained consistent and robust even after the 48 h incubation period. It is worth considering that collecting data from a single endpoint within a 48 h fermentation may not provide a comprehensive overview of the rumen prokaryotic microbiome (structure and composition) profile dynamics. To obtain a more precise and thorough understanding, it is advisable to include multiple time points of sampling in the fermentation. In general, this study confirms that the in vitro rumen fermentation technique allows for a high degree of species diversity that closely approximates the rumen in vivo.

**Author Contributions:** Conceptualization, R.D., R.S. and H.H.H.; methodology, R.D. and R.S.; software, R.D., A.L.A.N. and R.S.; validation, R.D., R.S. and H.H.H.; formal analysis, R.D. and R.S.; investigation, R.D. and R.S.; resources, R.D. and H.H.H.; data curation, R.D. and R.S.; writing—original draft preparation, R.D.; writing—review and editing, R.D., R.S., P.K., A.L.A.N. and H.H.H.; supervi-

sion, A.L.A.N. and H.H.H.; project administration, R.D. and H.H.H.; funding acquisition, R.D. and H.H.H. All authors have read and agreed to the published version of the manuscript.

**Funding:** This research received no external funding.

**Institutional Review Board Statement:** The cannulated animal use was authorized by Danish law under the research animal license no. 2012-15-2934-00648.

**Informed Consent Statement:** Not applicable.

**Data Availability Statement:** Data from the sequencing raw reads can be found in the NCBI SRA database under the accession number PRJNA1020745.

**Acknowledgments:** We would like to acknowledge lab technicians Anni Christiansen and Lotte Ørbæk for their contributions during the experimental procedures and other technical supports in the lab.

**Conflicts of Interest:** The authors declare that they have no competing interests.

**Abbreviations**

| | |
|---|---|
| ASV | Amplicon sequence variant |
| DADA2 | Deficiency of adenosine deaminase 2 |
| DNA | Deoxyribonucleic acid |
| IVRF | In vitro rumen fermentation |
| MS | Maize silage |
| PCoA | Principle coordinate analysis |
| PCR | Polymerase chain reaction |
| QIIME2 | Quantitative insights into microbial ecology 2 |
| RF | Rumen fluid |
| rRNA | Ribosomal ribonucleic acid |
| VFAs | Volatile fatty acids |

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
