# Peer review of "Prokaryote Composition and Structure of Rumen Fluid before and after In Vitro Rumen Fermentation"

_fermentation, doi:10.3390/fermentation10020108_

Round 1

Reviewer 1 Report

Comments and Suggestions for Authors

General comments:

It is well known that the efficiency of the microbially driven fermentation process in the rumen is closely related to the efficiency of the ruminant hosts. Moreover, recently there have been many scientific attempts to reveal conditions under which in vitro and in vivo experiments achieve similar and close final results. Such experiments deserve attention because in this way they can contribute to the new global goal of science without experimental animals. In other words, the article is current and of great interest, especially from a scientific point of view.

Overall, the introduction provides enough targeted information and all references included are relevant.

The applied methods, including in vitro fermentation, DNA extraction and bioinformatics, are described with high professionalism and are acceptable.

The obtained results are correct and very well presented, they are important, original, acceptable. They confirm the authors' hypothesis and may indeed be a step towards the global goal of science without experimental animals.

The discussion presented is very comprehensive and convincingly done. Special attention is paid to the results obtained for the significant differences between in vitro and in vivo at the level of Cyanobacteria, which differ from all others. This is very positive.

The conclusion is consistent with the evidence presented.

All cited literature sources (33) included in the article are relevant to the study.

Specific comment:

The title of the article "Bacterial composition and structure are similar in in vivo and in vitro rumen fluid" in this formulation reflects some of the conclusions. I believe it is more appropriate to change it to reveal the main content of the research done.

Regarding the chosen experimental design: it would be good to present strong arguments to the authors regarding their choice of this particular in vitro system - static instead of e.g. the semi-dynamic.

It is better to use italic for in vitro, in vivo, ad libitum during whole text. 

L. 132 - 23,350 ± 8561 or 23,350 ± 8,561?

L. 144 – “RF” has to be after “rumen fluid

L. 289 – correct cited is “R Core Team (2022). R: A Language and Environment for Statistical Computing. R Foundation for Statistical Computing”.

The abbreviations used is entered very properly. Once entered they can use one of the terms, not both (pl see l. 200).

Author Response

We are grateful for the insightful feedback provided, and we are pleased to inform you that we have addressed all of your comments. Your valuable input has significantly improved the quality of our research. Thank you for your time and expertise in helping us refine our work.

General comments:

It is well known that the efficiency of the microbially driven fermentation process in the rumen is closely related to the efficiency of the ruminant hosts. Moreover, recently there have been many scientific attempts to reveal conditions under which in vitro and in vivo experiments achieve similar and close final results. Such experiments deserve attention because in this way they can contribute to the new global goal of science without experimental animals. In other words, the article is current and of great interest, especially from a scientific point of view.

Overall, the introduction provides enough targeted information and all references included are relevant.

The applied methods, including in vitro fermentation, DNA extraction and bioinformatics, are described with high professionalism and are acceptable.

The obtained results are correct and very well presented, they are important, original, acceptable. They confirm the authors' hypothesis and may indeed be a step towards the global goal of science without experimental animals.

The discussion presented is very comprehensive and convincingly done. Special attention is paid to the results obtained for the significant differences between in vitro and in vivo at the level of Cyanobacteria, which differ from all others. This is very positive.

The conclusion is consistent with the evidence presented.

All cited literature sources (33) included in the article are relevant to the study.

Specific comment:

The title of the article "Bacterial composition and structure are similar in in vivo and in vitro rumen fluid" in this formulation reflects some of the conclusions. I believe it is more appropriate to change it to reveal the main content of the research done.

Corrected, we have changed the title.

Regarding the chosen experimental design: it would be good to present strong arguments to the authors regarding their choice of this particular in vitro system - static instead of e.g. the semi-dynamic.

Thank you for your suggestion. We have expanded the text in this section. However, this research does  not aim to compare between static and semi dynamic system and the comparison is out of the scope of this research.  

It is better to use italic for in vitro, in vivo, ad libitum during whole text. 

Corrected

  1. 132 - 23,350 ± 8561or 23,350 ± 8,561?

Corrected

  1. 144 – “RF”has to be after “rumen fluid

Corrected

  1. 289 – correct cited is “R Core Team (2022). R: A Language and Environment for Statistical Computing. R Foundation for Statistical Computing”.

Corrected

The abbreviations used is entered very properly. Once entered they can use one of the terms, not both (pl see l. 200).

Corrected

Reviewer 2 Report

Comments and Suggestions for Authors

This study aimed to investigate the impact of the IVRF on the microbiome before and after fermentation assays, which is an interesting startpoint. However, the major content is not same as the title " Bacterial composition and structure are similar in in-vivo and in-vitro rumen fluid". If only see the title, I thought the authors will compare the rumen microbiome difference after 48h maize silage of both in-vivo and in-vitro. But, the authors only focused on the pre- and post- in-vitro procedure. If compared animals consumed maize silage 48h rumen microbiome with 48h rumen microbiome after in-vitro, I will strongly agree its publication. Based on the current situation, a deep analysis or improvement of the content is necessary.

Using LeFse to identify the changed bacteria is necessary.

On figure 3, OTU may be wrong.

Comments on the Quality of English Language

English is good, but need to be improved.

Author Response

We are grateful for the insightful feedback provided, and we are pleased to inform you that we have addressed all of your comments. Your valuable input has significantly improved the quality of our research. Thank you for your time and expertise in helping us refine our work.

This study aimed to investigate the impact of the IVRF on the microbiome before and after fermentation assays, which is an interesting startpoint. However, the major content is not same as the title " Bacterial composition and structure are similar in in-vivo and in-vitro rumen fluid". If only see the title, I thought the authors will compare the rumen microbiome difference after 48h maize silage of both in-vivo and in-vitro. But, the authors only focused on the pre- and post- in-vitro procedure. If compared animals consumed maize silage 48h rumen microbiome with 48h rumen microbiome after in-vitro, I will strongly agree its publication. Based on the current situation, a deep analysis or improvement of the content is necessary.

We have changed the title to make it more clear

Using LeFse to identify the changed bacteria is necessary.

We have incorporated the LeFSe analysis

On figure 3, OTU may be wrong.

Corrected

Reviewer 3 Report

Comments and Suggestions for Authors

Reviewer’s comments on the manuscript by Zhang et al. entitled:  Bacterial composition and structure are similar in in-vivo and in-vitro rumen fluid.

Manuscript ID: fermentation-2826853

January 2024.

The manuscript is interesting topic, however, the author should be rewritten the experimental design, need to descript the material and methods detail!!!

Specific Comments are as follow:

in-vivo and in-vitro, italic, the same as the whole manuscript.

L10-13: delete these two sentences.

L79-91: I total did not understand the experimental design, the authors want to compare to in-vivo and in-vitro rumen fluid, where is in vivo ruminal fluid? I did not see this information in this manuscript. Please detail the information, I am very confuse, the description of this section is very confusing. In addition, please descript the in vitro and in vivo procedure detail!!!!!

Where is Statistical analysis section????

L160-161: Why the authors analysis post-48 h fermentation of maize silage??? Please explain the reason. In addition, I do not know how to compare? Do you compare fresh rumen fluid and maize silage incubation 48 h??? Which is reference? The authors mean fresh sample and after incubation sample, however, after incubation need to add substrate (maize silage), how about the fresh sample??

L165-166: Bacteroidota, Proteobacteria, Firmicutes, Verrucomicrobiota, Spirochaetota, Patescibacteria, Campilobacterota, and Cyanobacteria, italic, the same as the whole manuscript.

L212-223: I think one sentence could be summed up conclusion.

Author Response

We are grateful for the insightful feedback provided, and we are pleased to inform you that we have addressed all of your comments. Your valuable input has significantly improved the quality of our research. Thank you for your time and expertise in helping us refine our work.

Reviewer’s comments on the manuscript by Zhang et al. entitled:  Bacterial composition and structure are similar in in-vivo and in-vitro rumen fluid.

Manuscript ID: fermentation-2826853

January 2024.

The manuscript is interesting topic, however, the author should be rewritten the experimental design, need to descript the material and methods detail!!!

Thank you for your comments and suggestion, we have addressed all of your concerns. 

Specific Comments are as follow:

in-vivo and in-vitro, italic, the same as the whole manuscript.

Corrected.

L10-13: delete these two sentences.

Corrected.

L79-91: I total did not understand the experimental design, the authors want to compare to in-vivo and in-vitro rumen fluid, where is in vivo ruminal fluid? I did not see this information in this manuscript. Please detail the information, I am very confuse, the description of this section is very confusing. In addition, please descript the in vitro and in vivo procedure detail!!!!!

Corrected and thank you for highlighting it. Please see the new text in materials and method section.

Where is Statistical analysis section????

The statistical descriptions have been added in the  Bioinformatics section- Line 142-149. 

L160-161: Why the authors analysis post-48 h fermentation of maize silage??? Please explain the reason. In addition, I do not know how to compare? Do you compare fresh rumen fluid and maize silage incubation 48 h??? Which is reference? The authors mean fresh sample and after incubation sample, however, after incubation need to add substrate (maize silage), how about the fresh sample??

We analyzed post-48 h fermentation of maize silage because we aimed to characterize the rumen bacterial profile in pre- and post-fermentation samples. We compared the fresh rumen fluid (0 h, without maize silage; reference) with the same rumen fluid incubated with maize silage (substrate added) for 48h. Please see L185-188.

L165-166: Bacteroidota, Proteobacteria, Firmicutes, Verrucomicrobiota, Spirochaetota, Patescibacteria, Campilobacterota, and Cyanobacteria, italic, the same as the whole manuscript.

Corrected.

L212-223: I think one sentence could be summed up conclusion.

Thank you for your suggestion. While we appreciate the conciseness of summarizing the conclusion in one sentence, our aim is to ensure comprehensive coverage and clarity in our conclusions. Therefore, we prefer to maintain the current structure to provide readers with a thorough understanding of the research findings.

Round 2

Reviewer 2 Report

Comments and Suggestions for Authors

The revision is good. 

Reviewer 3 Report

Comments and Suggestions for Authors

I have no any question.